# Evaluation of the PSMA-Binding Ligand ^212^Pb-NG001 in Multicellular Tumour Spheroid and Mouse Models of Prostate Cancer

**DOI:** 10.3390/ijms22094815

**Published:** 2021-05-01

**Authors:** Vilde Yuli Stenberg, Roy Hartvig Larsen, Li-Wei Ma, Qian Peng, Petras Juzenas, Øyvind Sverre Bruland, Asta Juzeniene

**Affiliations:** 1Department of Radiation Biology, Institute for Cancer Research, The Norwegian Radium Hospital, Oslo University Hospital, 0379 Oslo, Norway; li-wei.ma@rr-research.no (L.-W.M.); asta.juzeniene@rr-research.no (A.J.); 2Department of Research and Development, Nucligen AS, 0379 Oslo, Norway; sciencons@gmail.com; 3Institute for Clinical Medicine, University of Oslo, 0318 Oslo, Norway; OSB@ous-hf.no; 4Department of Pathology, The Norwegian Radium Hospital, Oslo University Hospital, 0379 Oslo, Norway; qian.peng@rr-research.no (Q.P.); petras.juzenas@rr-research.no (P.J.); 5Department of Oncology, The Norwegian Radium Hospital, Oslo University Hospital, 0379 Oslo, Norway

**Keywords:** prostate-specific membrane antigen, metastatic castration-resistant prostate cancer, targeted alpha therapy, p-SCN-Bn-TCMC-PSMA ligand (NG001), ^212^Pb

## Abstract

Radioligand therapy targeting the prostate-specific membrane antigen (PSMA) is rapidly evolving as a promising treatment for metastatic castration-resistant prostate cancer. The PSMA-targeting ligand p-SCN-Bn-TCMC-PSMA (NG001) labelled with ^212^Pb efficiently targets PSMA-positive cells in vitro and in vivo. The aim of this preclinical study was to evaluate the therapeutic potential of ^212^Pb-NG001 in multicellular tumour spheroid and mouse models of prostate cancer. The cytotoxic effect of ^212^Pb-NG001 was tested in human prostate C4-2 spheroids. Biodistribution at various time points and therapeutic effects of different activities of the radioligand were investigated in male athymic nude mice bearing C4-2 tumours, while long-term toxicity was studied in immunocompetent BALB/c mice. The radioligand induced a selective cytotoxic effect in spheroids at activity concentrations of 3–10 kBq/mL. In mice, the radioligand accumulated rapidly in tumours and was retained over 24 h, while it rapidly cleared from nontargeted tissues. Treatment with 0.25, 0.30 or 0.40 MBq of ^212^Pb-NG001 significantly inhibited tumour growth and improved median survival with therapeutic indexes of 1.5, 2.3 and 2.7, respectively. In BALB/c mice, no signs of long-term radiation toxicity were observed at activities of 0.05 and 0.33 MBq. The obtained results warrant clinical studies to evaluate the biodistribution, therapeutic efficacy and toxicity of ^212^Pb-NG001.

## 1. Introduction

The majority of metastases (>85%) in metastatic castration-resistant prostate cancer (mCRPC) patients overexpress prostate-specific membrane antigen (PSMA), which is a promising target for radionuclide therapy [1,2]. In recent years, several small molecular PSMA-targeting ligands have been developed and used as tumour-targeting vectors for radionuclides [3,4,5]. The ^177^Lu-chelated PSMA-binding ligands PSMA-617 and PSMA-I&T (t_1/2_ = 6.7 days) have been extensively evaluated in clinical trials and have shown therapeutic efficacy and limited toxicity in mCRPC patients [6,7]. However, up to 30% of patients do not respond or develop resistance to ^177^Lu-PSMA therapy [8,9,10]. In these patients, treatment may be more effective when PSMA-617 is labelled with the alpha emitter ^225^Ac (t_1/2_ = 10.0 days) instead of the beta emitter ^177^Lu [11,12,13,14]. The improved antitumour response of such targeted alpha therapy (TAT) is due to the high linear energy transfer (LET, ~100 keV/µm) of alpha particles, which leads to highly effective cell killing via irreparable DNA breaks. In contrast, low-LET beta particles (~0.3 keV/µm) produce individual DNA lesions that are more easily reparable. The high LET and short range of alpha emitters (<100 µm) makes them particularly suitable for treatment of micrometastases and single-cell disease due to the more efficient and specific killing of tumour cells while sparing surrounding normal tissue [11,15,16]. TAT using ^213^Bi-labelled PSMA-617 has also shown considerable antitumour activity in a mCRPC patient [17]. A recently published review provides a comprehensive overview of the preclinical and clinical status of PSMA-TAT for mCRPC [18]. However, several limitations are associated with the use of ^225^Ac and ^213^Bi, including availability, logistics related to the short half-life of ^213^Bi (t_1/2_ = 46 min) and toxicity in salivary glands of ^225^Ac [12,19]. In addition, the alpha-emitting daughters of ^225^Ac may dissociate from the radioligand during decay, possibly followed by accumulation in healthy tissues such as the kidneys [20]. Tandem therapy with low-activity ^225^Ac-PSMA-617 plus full-activity ^177^Lu-PSMA-617 is currently under clinical investigation to increase efficacy and reduce toxicity of ^225^Ac [21,22].

Another suitable radionuclide for PSMA-TAT is the beta emitter ^212^Pb (t_1/2_ = 10.6 h), which acts as an in vivo generator for its alpha-emitting progenies ^212^Bi (t_1/2_ = 60.6 min) and ^212^Po (t_1/2_ = 0.3 microseconds), yielding on average one alpha particle per ^212^Pb atom. Radioimmunotherapy with ^212^Pb-TCMC-trastuzumab has been clinically tested against ovarian cancer and demonstrated signs of antitumour effect without any reported safety issues [23]. Several research groups have developed PSMA-targeting ligands specifically designed for ^212^Pb chelation that show promising tumour-targeting in preclinical models [5,24,25,26]. Recently, we developed and compared the glutamate-urea-based PSMA-targeting ligand p-SCN-Bn-TCMC-PSMA (NG001) with PSMA-617 [25,26]. Both ligands were rapidly labelled with ^212^Pb using a liquid ^224^Ra/^212^Pb-generator solution, and the resulting radioligands efficiently targeted PSMA-positive cells in vitro and in vivo. In the present study, the biodistribution of ^212^Pb-NG001 was investigated at several time points post injection. We aimed to evaluate the therapeutic efficacy of ^212^Pb-NG001 both in vitro in a C4-2 multicellular spheroid model and in vivo in athymic nude mice bearing C4-2 prostate cancer xenografts. In addition, the long-term toxic effects of the radioligand were studied in immunocompetent BALB/c mice.

## 2. Results

### 2.1. Toxicity of ^212^Pb-NG001 in C4-2 Multicellular Spheroid Model

Each spheroid was formed from 500 cells and reached a diameter of 400 ± 60 µm (volume of 26 ± 3 µm^3^) after 5 days in culture (day of treatment, day 0). The formation of a necrotic core was not observed in the spheroids at this stage (Appendix A), indicating that all cells were metabolically active. Compared with spheroids preblocked with an excess of unlabelled ligand, specific growth delay of nonblocked spheroids was observed at activity concentrations of 5–10 kBq/mL of ^212^Pb-NG001 after 4 h of incubation and at 3–10 kBq/mL after 48 h (Figure 1). The doubling time of the spheroids was 6 days for the control and increased to 9 and 17 days for spheroids treated with 5 and 10 kBq/mL of the radioligand for 4 h, respectively. For the spheroids treated for 48 h, the doubling time increased to 17–22 days for activity concentrations of 3–10 kBq/mL.

### 2.2. Biodistribution of ^212^Pb-NG001 in Mice with C4-2 Xenografts

The effective half-lives of ^212^Pb-NG001 in selected organs/tissues were obtained by fitting exponential functions to the biodistribution data (Bq/g vs. time), and the biological half-lives were then calculated. The estimated half-lives are presented in Table 1 (calculated from data in Appendix A).

The radioligand showed fast clearance from blood (Table 1 and Appendix A). One hour after injection, less than 2.5 %ID/g of ^212^Pb-NG001 was present in blood and other nontargeted tissues, including liver, spleen, intestines, skin, femur, skull and salivary glands (Figure 2). The radioligand was excreted mainly via the renal pathway, yielding high initial levels in urine and kidneys. However, radioactivity was rapidly cleared from the kidneys (Table 1), with radioactivity being reduced more than 6-fold within 4 h and reduced to below 5.3 %ID/g at 24 h post injection (Figure 2).

In C4-2 tumour-bearing mice, ^212^Pb-NG001 showed a fast tumour accumulation, already reaching 23.3 ± 8.7 %ID/g at 1 h post injection (Figure 2). Clearance from the tumours was significant from 1 to 2 h but was slow thereafter (Table 1 and Figure 2). Radioactivity levels in the tumours remained above 10 %ID/g after 24 h, resulting in improved tumour-to-kidney ratios over time. No significant correlation between tumour size and uptake of ^212^Pb-NG001 was observed in tumours up to 1.1 g (Appendix A). However, a trend of increased radioligand uptake in tumours of smaller size was observed at 1 h post injection.

Activity values for ^212^Pb and ^212^Bi for the radioligand were similar in each tissue during the 24 h study period (Appendix A), indicating no extensive translocation of the ^212^Bi daughter from ^212^Pb in the studied organs.

Very low tumour uptake of ^212^Pb-NG001 was observed in mice bearing PSMA-negative PC-3 xenografts after 1 and 4 h (<1.3 %ID/g, N = 2–3, results not shown), indicating that the tumour uptake was strongly related to PSMA expression. Nonchelated cationic ^212^Pb^2+^, injected as a control, was detected in kidneys (34.2–49.0 %ID/g), liver (8.3–9.1 %ID/g) and bones (7.3–10.4 %ID/g at 1–4 h, 1.0–2.6 %ID/g at 8 h) at 1–8 h post injection (Appendix A). The difference in biodistribution of free ^212^Pb compared to ^212^Pb-NG001 (Figure 2 and Appendix A) indicates appropriate in vivo stability of the radioligand.

### 2.3. Therapeutic Effect of Radioligand in Athymic Nude Mice with C4-2 Xenografts

The therapeutic effect of 0.25, 0.30 and 0.40 MBq of ^212^Pb-NG001 was assessed in athymic nude mice bearing C4-2 tumours after intravenous administration of a single injection dose in three independent experiments. Mice in the control groups showed rapid tumour growth, resulting in median survivals of only 16.5, 23 and 27 days (Table 2). Tumour growth delay was observed for mice treated with 0.25, 0.30 and 0.40 MBq of ^212^Pb-NG001, leading to significantly improved median survival with therapeutic indexes (TI) of 1.5, 2.3 and 2.7, respectively (Table 2; Figure 3, Appendix A).

All control mice bearing C4-2 tumours lost 5–10% of body weight, which was not prevented by the treatment with ^212^Pb-NG001 (Appendix A). Observations of the liver, kidneys and spleen during autopsy, and their respective weights, did not reveal any abnormalities for the treated mice (*p* > 0.05 for the treatment groups compared to the control, Appendix A). Haematological analysis at the day of sacrifice showed no difference in white blood cell, red blood cell, haemoglobin or platelet counts between the control and the treatment groups (*p* > 0.05, Appendix A). Furthermore, neither changes in serum glutamic oxaloacetic transaminase (GOT), glutamic pyruvic transaminase (GPT), alkaline phosphatase (ALP) and bilirubin values for liver function nor urea and creatine values for renal function were observed (*p* > 0.05, Appendix A). These findings suggest that the used activities did not induce any significant toxicity in the short to medium time period (up to 28 weeks) after treatment.

### 2.4. Long-Term Toxicity Evaluation of ^212^Pb-NG001 in BALB/C Mice

Long-term effects of ^212^Pb-NG001 were assessed in immunocompetent BALB/c mice with a 12-month follow-up. Changes in body weight and blood parameters after administration of 0.05 and 0.33 MBq of the radioligand are presented in Figure 4. Body weight gradually increased in all groups, and no significant difference between the control and treatment groups was observed (*p* > 0.05). Throughout the study, blood parameters were similar in the control and treatment groups, and all parameters were within the reference range.

Histology of the kidneys, testes, salivary glands (serous and mucinous glands), liver, spleen and femur did not reveal any morphological abnormalities for the treated mice (Figure 5 and Figure 6, results not shown for the lowest activity). No histological alterations of glomeruli (capillaries and Bowman’s capsule), tubules (proximal/distal convoluted tubules and collecting ducts), interstitium (no inflammation or fibrosis) or blood vessels (no congestion and haemorrhage) of the kidneys were observed (Figure 5).

Some congestion of the spleen and slight degeneration of some liver cells (vacuoles) were observed in all groups, possibly explained by old age, manner of death or method of euthanasia. Serum parameters and weights of the kidneys, liver and spleen in the treatment groups were similar to those in the control (*p* > 0.05, Figure 7 and Appendix A). These findings suggest that 0.05–0.33 MBq of ^212^Pb-NG001 do not induce significant long-term radiation toxicity.

## 3. Discussion

In the present study, ^212^Pb-NG001 demonstrated PSMA-specific cancer cell targeting and tumour growth delay in multicellular spheroid and mouse models of prostate cancer. No long-term radiation toxicity was observed in immunocompetent mice receiving therapeutic levels of the radioligand.

Cytotoxicity was assessed in multicellular C4-2 spheroids (diameter ~400 µm) that resemble micrometastases and small avascular tumours [27]. The ^212^Pb-NG001 induced growth inhibition at activity concentrations of 3–10 kBq/mL (Figure 1), corresponding to 15–50 MBq per patient (~5 L blood), which is similar to the administered doses of ^212^Pb-TCMC-trastuzumab in clinical trials (13–49 MBq) [23,28]. The findings indicate that ^212^Pb-NG001 can penetrate and induce cytotoxic effect in microscopic and nonvascularised metastases at clinically relevant doses. In the spheroid experiment, a relevant specific activity of 0.514 MBq/nmol of ^212^Pb-NG001 was used as specific activity can significantly influence the effect of alpha-emitting radioligands [29].

Compared to other PSMA-binding ligands, the tumour uptake of ^212^Pb-NG001 after 1 h was similar to that of ^203^Pb-L2 but considerably higher than that reported for ^203^Pb-CA012 and for most studies of ^177^Lu-PSMA-617 (Figure 2 and Table 3) [5,24,30,31,32]. In a study by Benesova et al. [33], tumour uptake of ^177^Lu-PSMA-617 was around three times higher after 1, 4 and 24 h compared to other studies of the same radioligand (Kuo et al. [30], Benesova et al. [31] and Kelly et al. [32]). However, direct comparison of PSMA-binding ligands must be made carefully as different mouse strains and cell lines were used in the various studies. For example, PC-3 PIP cells, which were used in the study reported by Banerjee et al. [24] and Benesova et al. [33], have around two times higher PSMA expression than C4-2 and LNCaP cells, allowing more radioligand to bind specifically to the PC-3 PIP cells [34,35].

After clearance from 1 to 2 h after injection (from 23.3 to 17.6 %ID/g), ^212^Pb-NG001 showed long retention in the tumours with uptake values remaining above 10 %ID/g at 24 h post injection (Table 3), corresponding to two physical half-lives of the ^212^Pb radionuclide. In contrast, several studies have reported a strong reduction in tumour levels of ^177^Lu-PSMA-617 before the physical half-life of 6.7 days is reached (levels of only 0.7 %ID/g after 4 days by Fendler et al. [36], of 3.5 %ID/g after 4 days by Kelly et al. [32] and of 7 %ID/g after 5 days by Kuo et al. [30]). Consequently, the biological half-life of the radioligand in the tumour is considerably shorter than the physical half-life of the radionuclide, and ^177^Lu delivers a large amount of its radiation after it has cleared from the tumour tissue. Therefore, ^212^Pb seems more suitable for ligands with such rapid pharmacokinetics as much of the radioactive decay occurs when the level of the radioligand in the tumour is high. This indicates that the biological half-life of the radioligand in tumour (Table 1) matches the physical half-life of ^212^Pb quite well. Furthermore, shorter-lived nuclides, such as ^212^Pb, have a higher initial dose rate compared to longer-lived nuclides, such as ^177^Lu and ^225^Ac, i.e., ^212^Pb delivers its radiation dose in a shorter time span [37]. Radiation with high dose rates might be more effective than the same dose delivered with low dose rates as low dose rates provide a greater time interval for DNA damage repair and cell proliferation to occur during irradiation [38].

A trend of increased radioligand uptake (%ID/g) was observed in smaller tumours compared to larger tumours at 1 h post injection (Appendix A). A possible explanation is that smaller tumours have denser microvasculature [39], and hence the tumour binding sites are more readily accessible for circulating radioligands in the distribution phase. The tumour uptake of PSMA-targeted radioligands is also correlated with the degree of PSMA expression and the fraction of PSMA-positive cells [34]. In smaller tumours, PSMA-positive prostate cells constitute a larger portion of the tumour compared to larger tumours, which contain more normal stromal and epithelial cells. At later time points (>1 h), uptake values were similar in tumours of different size, suggesting that tumour cell internalization retains the radioligand to the same extent in both small and large tumours.

Low molecular ligands are mainly cleared via the renal pathway [40,41], resulting in high initial activity levels in the kidneys for all radioligands (Table 3). However, clearance from the kidneys was fast. Compared to the maximum uptake in the kidneys at 1 h post injection, there was 65–85% reduction in activity after 4 h and 82–99% reduction after 24 h for all radioligands. The kidney uptakes (1–4 h) of ^212^Pb-NG001, ^203^Pb-CA012 and ^203^Pb-L2 were considerably lower than that of ^177^Lu-PSMA-617 in most studies (Table 3). The lower kidney uptake of ^177^Lu-PSMA-617 in the study conducted by Benesova et al. [33] can be explained by the tumour sink effect, where higher tumour load (higher PSMA expression of PC-3 PIP tumours) lead to lower kidney dose [42]. A low kidney uptake of ^212^Pb-labelled radioligands is critical because of the higher dose rate of ^212^Pb that could otherwise represent a potential toxicity problem. It is noteworthy that the kidney retention of PSMA-binding ligands in humans is expected to be significantly lower than in mice as human kidneys have lower expression of PSMA receptors [43,44]. Therefore, the human kidney dose cannot be directly extrapolated from mouse kidney uptake. However, kidney toxicity may be a potential problem in clinical use of small molecular PSMA radioligands and should be carefully monitored [45]. The standard strategy to avoid kidney damage in clinical targeted radionuclide therapy is to ensure that the patient is well hydrated and to co-infuse the basic amino acids lysine and arginine to reduce renal retention of the radioligands [41,46,47]. In addition, several strategies to reduce kidney toxicity are currently under investigation, including dose fractionation and co-injection with cold PSMA ligand, succinylated gelatin or radioprotectors, such as α1-microglobulin or amifostine [41,48,49,50,51,52,53].

The large difference in biodistribution of ^212^Pb-NG001 and free cationic ^212^Pb^2+^ (Figure 2 and Appendix A) indicates a relevant in vivo stability of ^212^Pb-NG001. However, a possible challenge with the use of alpha-emitting radionuclides is a poor retention of daughter nuclides in the chelator after decay [20]. For ^212^Pb-labelled radioligands, around 30% of progeny ^212^Bi could dissociate from S-2-(4-isothiocyanatobenzyl)-1,4,7,10-tetraazacyclododecane tetraacetic acid (DOTA) and S-2-(4-isothiocyanatobenzyl)-1,4,7,10-tetraaza-1,4,7,10-tetra(2-carbamoylmethyl)-cyclododecane (TCMC) chelators of ligand complexes [20,54,55]. The TCMC chelator of NG001 has all four chelator arms available for chelating purposes, compared to three available chelator arms in the DOTA chelator of PSMA-617, possibly increasing stability of the radionuclide–chelator complex [25]. This corresponds to findings by Maaland et al. that showed release of only 16% of ^212^Bi from the backbone-linked TCMC chelator of an anti-CD37 radioimmunoconjugate (^212^Pb-NNV003) [56]. Moreover, no extensive translocation of the ^212^Bi daughter from ^212^Pb-NG001 was detected in nontargeted organs during a 24 h study period in the current study (Appendix A), which is in accordance with a rapid clearance of the radioligand from blood and normal tissues and a strong retention of the radionuclide and progeny after tumour cell internalization [25,57]. For ^225^Ac-labelled radioligands, the challenge of recoiling alpha-emitting daughters is potentially even larger as ^225^Ac emits four alpha particles per decay compared to one alpha particle of ^212^Pb [20].

Treatment with 0.25–0.40 MBq of ^212^Pb-NG001 significantly inhibited tumour growth and improved survival in vivo (Table 2). The therapeutic effects of other PSMA-targeting ligands have been investigated in preclinical xenograft models (Table 4 and Appendix A). The ^212^Pb-L2 significantly inhibited tumour growth with TIs of 1.9 and 3.0 at activity doses of 1.5 and 3.7 MBq, respectively [24]. This is similar to the TIs of much lower injected activities of 0.25–0.40 MBq of ^212^Pb-NG001 in the current study (Table 4).

Therapeutic evaluation of ^177^Lu-PSMA-617 in vivo is well reported in the literature (Table 4 and Appendix A), but large discrepancies were observed in the various studies. As mentioned above, the selected cell lines have different PSMA expression and thus different numbers of available PSMA binding sites; PC-3 PIP cells have higher expression than C4-2 and LNCaP cells, which in turn have higher expression than RM1-PGLS and LS174T-PSMA cells [36,44]. In addition, the radiation sensitivity of the mouse model, growth pattern of the cell line and tumour size at the time of treatment must be taken into consideration. SCID and NOD-SCID gamma mice are more radiation sensitive than NOD rag gamma and athymic nude mice, which may affect the radiation response. Less repair capacity against DNA damage in radiosensitive strains could result in prolonged tumour growth delay after radiation exposure [62,63,64]. Consequently, direct comparison of survival and tumour growth of different radioligands in various tumour models must be made carefully.

In studies with C4-2 tumour-bearing mice, activities of 15 and 30 MBq of ^177^Lu-PSMA-617 improved median survival with TIs of 1.7 and 3.1 (radiosensitive models, Appendix A) [60,61], similar to TIs of 1.5–2.7 for 0.25–0.40 MBq of ^212^Pb-NG001 in the present study (radioresistant model). Comparable TIs for ^212^Pb-NG001 as reported for ^177^Lu-PSMA-617 in tumour-bearing mice (at activities of a factor of 200–267 from clinically relevant doses, Table 5) indicates a similar ability to target macroscopic tumours. In addition, alpha-generating ^212^Pb-NG001 will have the potential to target microscopic metastases that are not efficiently irradiated with beta-emitting ^177^Lu-PSMA-617 [16,24].

TAT with ^225^Ac-PSMA-617 demonstrated tumour growth inhibition at activities of 0.02, 0.04 and 0.10 MBq in a radiosensitive model (Appendix A) [59] However, mice treated with 0.10 MBq experienced toxicity, including severe weight loss, while mice treated with 0.02 and 0.04 MBq experienced transient weight loss. In a more radioresistant model, 0.03 MBq of ^225^Ac-PSMA-617 produced only marginal antitumour responses (TI of 1.1) and showed no signs of toxicity [61].

The long-term follow up of immunocompetent BALB/c mice treated with 0.05 and 0.33 MBq (0.16 + 0.17 MBq) of ^212^Pb-NG001 showed no signs of radiation toxicity (Figure 4, Figure 5, Figure 6 and Figure 7 and Appendix A). This is in accordance with a study conducted by Banerjee et al. [24], where no changes in body weight or serum parameters were detected 12 months after administration of activity doses below 0.74 MBq of ^212^Pb-L2 in immunocompetent CD-1 mice. However, the activity dose of ^212^Pb-L2 needed to gain a therapeutic effect in a xenograft model (1.5 MBq, TI of 1.9, Appendix A) induced mild changes in the cortical tubules of the kidneys after 12 months. In the present study, 0.33 MBq of ^212^Pb-NG001, an activity level sufficient for antitumour effect in a xenograft model (TI of 2.3, Table 2), induced no histopathologic abnormalities in the kidneys, testes, salivary glands, liver, spleen or femur 12 months after administration in immunocompetent BALB/c mice (Figure 5 and Figure 6). Furthermore, no observed changes in kidney weight or in the serum urea or creatinine parameters for renal function (Figure 7 and Appendix A) suggest no long-term renal toxicity despite the high initial kidney uptake of the radioligand (Figure 2).

## 4. Materials and Methods

### 4.1. Preparation of ^212^Pb and Activity Measurements

Radium-224 was extracted from a generator column containing DIPEX^®^ actinidine resin (Eichrom Technologies, Lisle, Illinois, USA) with immobilized ^228^Th (Eckert & Ziegler, Braunschweig, Germany) by eluting with 1 M HCl, as described previously [55,66]. Radioactive samples were measured on a Hidex Automatic Gamma counter (Hidex Oy, Turku, Finland) with 60–110 keV counting window for the determination of ^212^Pb activity and 520–640 keV window for the determination of ^212^Bi activity indirectly from the highly abundant ^208^Tl gamma radiation. To quantify ^212^Bi, samples were measured after approximately 1 h, when transient equilibrium between ^208^Tl and the ^212^Bi parent had been established but before ^212^Pb and ^212^Bi had reached equilibrium in case of significant translocation. A Capintec CRC-25R radioisotope dose calibrator (Capintec Inc., Ramsey, NJ, USA) was used during the radiolabelling procedures with calibration number setting of 667 for ^212^Pb in equilibrium with daughters [67].

### 4.2. Radiolabelling and Purification of PSMA Ligand

The PSMA ligand NG001 was supplied as HPLC purified and dried trifluoroacetic acid salt (purity of ≥98%) by MedKoo Biosciences (Morrisville, NC, USA). NG001 dissolved in 0.5 M ammonium acetate in 0.1 M HCl was labelled with ^212^Pb using a liquid ^224^Ra/^212^Pb generator solution, as previously described [25,68]. The ^212^Pb-NG001 was purified using PD Minitrap G-10 columns prepacked with Sephadex G-10 resin (GE Healthcare Bio-Sciences AB, Uppsala, Sweden) to remove free ^224^Ra to a level below 0.8%. The radiochemical purity (RCP) of the radioligand was measured by thin-layer chromatography, and radioligands with RCP >95% were used for the experiments.

### 4.3. Cell Line and Cell Binding Assay

The human prostate cancer cell lines C4-2 (PSMA-positive, ATCC^®^ CRL-3314™) [69] and PC-3 (PSMA-negative, ATCC^®^ CRL-1435™) [70] were obtained from ATCC (Manassas, Virginia). Cells were grown in RPMI 1640 medium (Merck Norge, Oslo, Norway) supplemented with 10% heat-inactivated foetal bovine serum (GE Healthcare Life Sciences, Chicago, IL, USA) and 100 units/mL penicillin and 100 µg/mL streptomycin (Sigma-Aldrich) at 37 °C in a humid atmosphere of 95% air and 5% CO_2_. Before injection of radiolabelled ligands in mice, binding of the radioligands was verified by measuring cell binding ability in C4-2 cells, as described previously [25].

### 4.4. Toxicity of Radioligand in a C4-2 Multicellular Spheroid Model

C4-2 spheroids were generated by cultivation of cells in liquid overlay in 1.5% agarose-coated flat bottom 96-well plates [71]. Cell suspensions of 500 cells in 100 µL medium were added to each well, followed by centrifugation of the plates at 470× *g* for 15 min. After an initial incubation time of 5 days, spheroids with diameter of ~400 µm were formed. The viability of the cells in the spheroids were measured by a fluorescence-based live–dead assay using fluorescein diacetate for live cells (FDA, Merck Norge) and propidium iodide for dead cells (PI, Merck Norge). Fluorescent images were taken by an inverted Axiovert 200M microscope (Carl Zeiss AG, Jena, Germany) with AxioVision Rel. 4.8 software (Carl Zeiss AG). Spheroids were then incubated with 1–10 kBq/mL of ^212^Pb-NG001 (specific activity of 0.514 MBq/nmol, 6 spheroids per group) for 4 or 48 h at 37 °C. Toxicity from nonspecifically bound radioligand was measured by blocking spheroids with an excess of unlabelled NG001 for 30 min before addition of the radioligand to occupy PSMA binding sites. After incubation, spheroids were washed 6 times with fresh medium and further incubated for 20 days with change of culture medium three times per week. Spheroid growth was evaluated two times per week using a bright-field microscope.

### 4.5. Animals and Tumour Xenografts

For the biodistribution and therapy studies, 105 male Hsd:Athymic Nude-Foxn1^nu^ mice bred at the Department of Comparative Medicine at the Norwegian Radium Hospital (Oslo University Hospital, Oslo, Norway) were used. For the toxicity study, 13 male BALB/cJRj mice from Janvier Labs (Le Genest-Saint-Isle, France) were used. The studies were approved by the Institutional Committee on Research Animal Care (Department of Comparative Medicine, Oslo University Hospital) and the Norwegian Food Safety Authority (Brumunddal, Norway, approval: FOTS ID 22197, 20/49651). All procedures and experiments involving animals in this study were performed in accordance with the Interdisciplinary Principles and Guidelines for the Use of Animals in Research, Marketing and Education (New York Academy of Sciences, New York, USA) and the EU Directive 2010/63/EU for animal experiments as well as the ARRIVE guidelines. The animals were maintained under specific pathogen-free conditions with ad libitum access to food and water. Cages (1–5 mice per cage) were housed in a scantainer, which was maintained at constant temperature (24 °C) and humidity (60%). The mice were around 4–6 weeks in age and weighed 25–35 g at the start of the study.

Mice were inoculated subcutaneously in both flanks with 10 × 10^6^ C4-2 cells in supplement-free RPMI1640 medium mixed 1:1 with Matrigel Matrix (Corning, NY, USA) in a total volume of 200 µL. In one of the biodistribution experiments, mice were inoculated with 5 × 10^6^ PC-3 cells (200 µL, 1:1 medium:Matrigel Matrix) as a negative control. The tumours were allowed to grow to reach a volume of 300–1500 mm^3^ for the biodistribution studies and a volume of 30–300 mm^3^ for the therapy studies, and tumour-bearing mice were randomised based on tumour size before radioligand injection. Mice were monitored 2–3 times per week for changes in body weight and tumour size and were euthanised by cervical dislocation if any of the following humane end points were reached: >20% weight loss from initial body weight, rapid weight loss (>10% within 2 days), tumours exceed 20 mm in any direction, ulcerate or interfere with normal behaviour, or any signs of severe sickness or discomfort.

### 4.6. Biodistribution of Radioligand in Mice with C4-2 Xenografts

Mice (8-12 weeks in age) were injected intravenously via tail vein with 6.6–58.6 kBq (0.06–0.6 nmol) of ^212^Pb-NG001 or 40–50 kBq of free ^212^Pb. At various time points after injection (from 15 min up to 48 h), organs/tissues were harvested, and radioligand uptake was calculated as a percentage of injected dose per gram tissue (%ID/g) [25]. Effective half-lives in selected organs were obtained by fitting monoexponential functions to the biodistribution data (Bq/g vs. time), normalised to 1 MBq/kg,
(1)$$y\left(t\right)=a\times e^{-\frac{\ln\left(2\right)\times t}{T_{\mathrm{effective}}}}$$

The biological half-lives were then calculated by the following formula:(2)$$\frac{1}{T_{\mathrm{effective}}}=\frac{1}{T_{\mathrm{biological}}}+\frac{1}{T_{\mathrm{physical}}}$$

### 4.7. Therapeutic Effect of Radioligand in Athymic Nude Mice with C4-2 Xenografts

The therapeutic effects of radioligands were studied in three independent studies. Mice were injected intravenously via tail vein with 0.25, 0.30 or 0.40 MBq of ^212^Pb-NG001 (0.17–0.50 nmol) in a volume of 100–150 µL. Control mice received 100–150 µL of 0.9% NaCl. At the day of sacrifice, blood was collected from the saphenous vein, and complete blood count was obtained using a haematology analyser (scil Vet abs, Kruuse, Drøbak, Norge). Blood was taken by cardiac puncture under gas anaesthesia (~3.5% sevoflurane (Baxter, Illinois) in oxygen at a flow rate of 0.5 L/min), and serum was collected and analysed for GOT, GPT, ALP, bilirubin, urea and creatinine (Reflotron Plus, Roche Diagnostics AS, Oslo, Norway). Gross observations of selected organs (liver, spleen and kidneys) were performed, and the organs were harvested and weighed.

### 4.8. Long-Term Toxicity of ^212^Pb-NG001 in BALB/C Mice

Mice were injected intravenously via tail vein with 0.05 MBq (0.13 nmol, 100 µL) or 0.33 MBq (0.73 nmol, 250 µL) of ^212^Pb-NG001. The high activity group was injected twice, one week apart (0.16 MBq in 100 µL + 0.17 MBq in 150 µL). Control mice received 0.9% NaCl (100 µL + 100 µL, one week apart). Blood was collected from the saphenous vein once per month. After 12 months, mice were sacrificed, and blood was collected by cardiac puncture for clinical chemistry. Selected organs (liver, spleen, kidney, femur, salivary glands and testes) were harvested and fixed in 4% formaldehyde in phosphate buffer (pH of ~7) for histopathological analysis. Fixed tissue samples were paraffin-embedded, and 3 µm sections were stained with haematoxylin and eosine (Dako, Agilent, CA, USA) before examination by a pathologist. Histological images were taken using a bright-field microscope (BX45, Olympus, Tokyo, Japan) with AxioVision Rel. 4.9.1 software (Carl Zeiss AG).

### 4.9. Statistics

The data sets were analysed for significance using a one-way ANOVA with multiple comparisons using SigmaPlot 14.0 software (Systat Software, Inc., San Jose, CA, USA). In the therapy studies, survival of the different treatment groups was estimated using the Kaplan–Meier survival analysis and compared using the log-rank test by pairwise comparisons (SigmaPlot). A *p*-value of <0.05 was considered statistically significant.

## 5. Conclusions

The presented results demonstrate that ^212^Pb-NG001 is a promising candidate for PSMA-targeting radioligand therapy of mCRPC. It has rapid pharmacokinetics with biological half-life in tumours that matches the physical half-life of the ^212^Pb nuclide, a desirable feature for successful PSMA-TAT. The ^212^Pb-NG001 induced cytotoxicity in a multicellular spheroid model at clinically relevant doses. In tumour-bearing mice, the radioligand showed high tumour uptake and retention as well as low uptake in nontargeted tissues. Injection doses of 0.25, 0.30 and 0.40 MBq of ^212^Pb-NG001 reduced tumour growth and improved the median survival with therapeutic indexes of 1.5, 2.3 and 2.7, respectively. In immunocompetent mice, 0.05 and 0.33 MBq of the radioligand induced no long-term toxicity as measured by changes in body weight, blood parameters and histopathologic evaluation of selected organs. These results warrant further exploration in early-phase clinical studies to evaluate the therapeutic efficacy and toxicity of ^212^Pb-NG001.

## Figures and Tables

**Figure 1 ijms-22-04815-f001:**
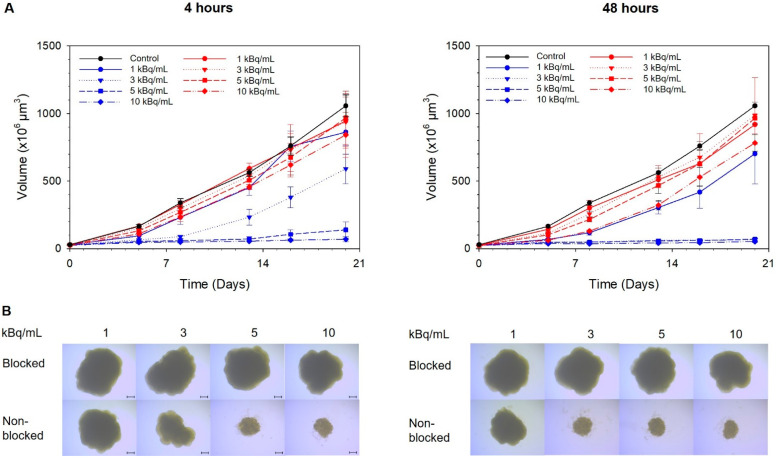
The influence of ^212^Pb-NG001 on C4-2 spheroid growth after incubation for 4 (left) or 48 (right) h at 37 °C. (**A**) Growth of spheroids in the nonblocked ^212^Pb-NG001 (blue), blocked ^212^Pb-NG001 (red) and control (black) groups was measured for up to 20 days and is presented as volume (×10^6^ µm^3^) ± SD (one experiment, N = 6 spheroids per group). The blocked group was preblocked with an excess of unlabelled NG001 to occupy PSMA binding sites. (**B**) Microscope images (×4 magnification) were taken at the predefined study end point of 20 days using a bright-field microscope with AxioVision Rel. 4.8 software; scale bars are 200 µm.

**Figure 2 ijms-22-04815-f002:**
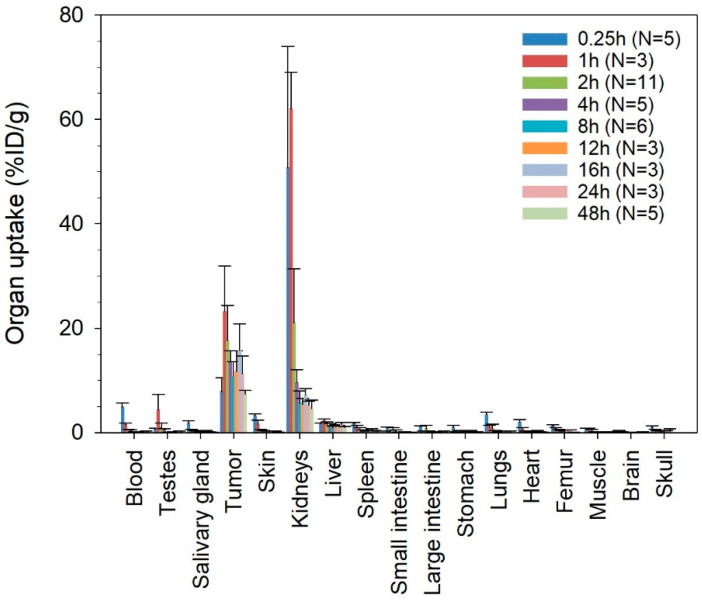
Decay-corrected percentage of injected activity per gram of tissue (%ID/g) ± SD of ^212^Pb-NG001 in athymic nude mice bearing human prostate C4-2 cancer xenografts at various time points after injection. N, number of mice per group.

**Figure 3 ijms-22-04815-f003:**
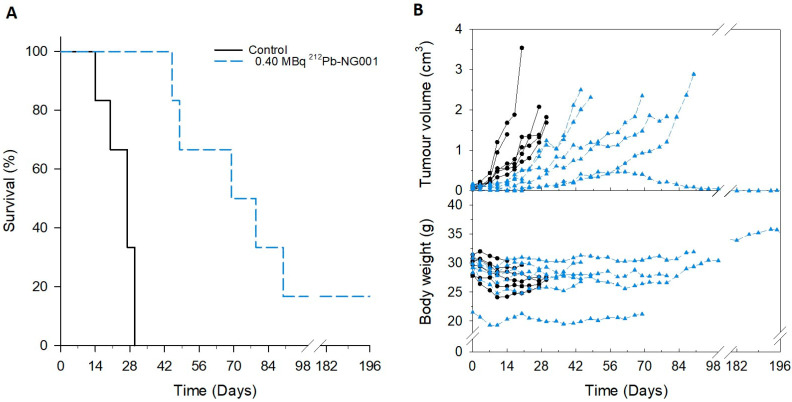
Survival (**A**) and changes in tumour volume and body weight (**B**) of athymic nude mice bearing prostate C4-2 tumours treated with saline or 0.40 MBq of ^212^Pb -NG001. Survival was estimated by Kaplan–Meier survival analysis followed by the log-rank test by pairwise comparisons. In (**B**), each line represents one mouse (only the largest tumour is shown), N = 6 mice per group.

**Figure 4 ijms-22-04815-f004:**
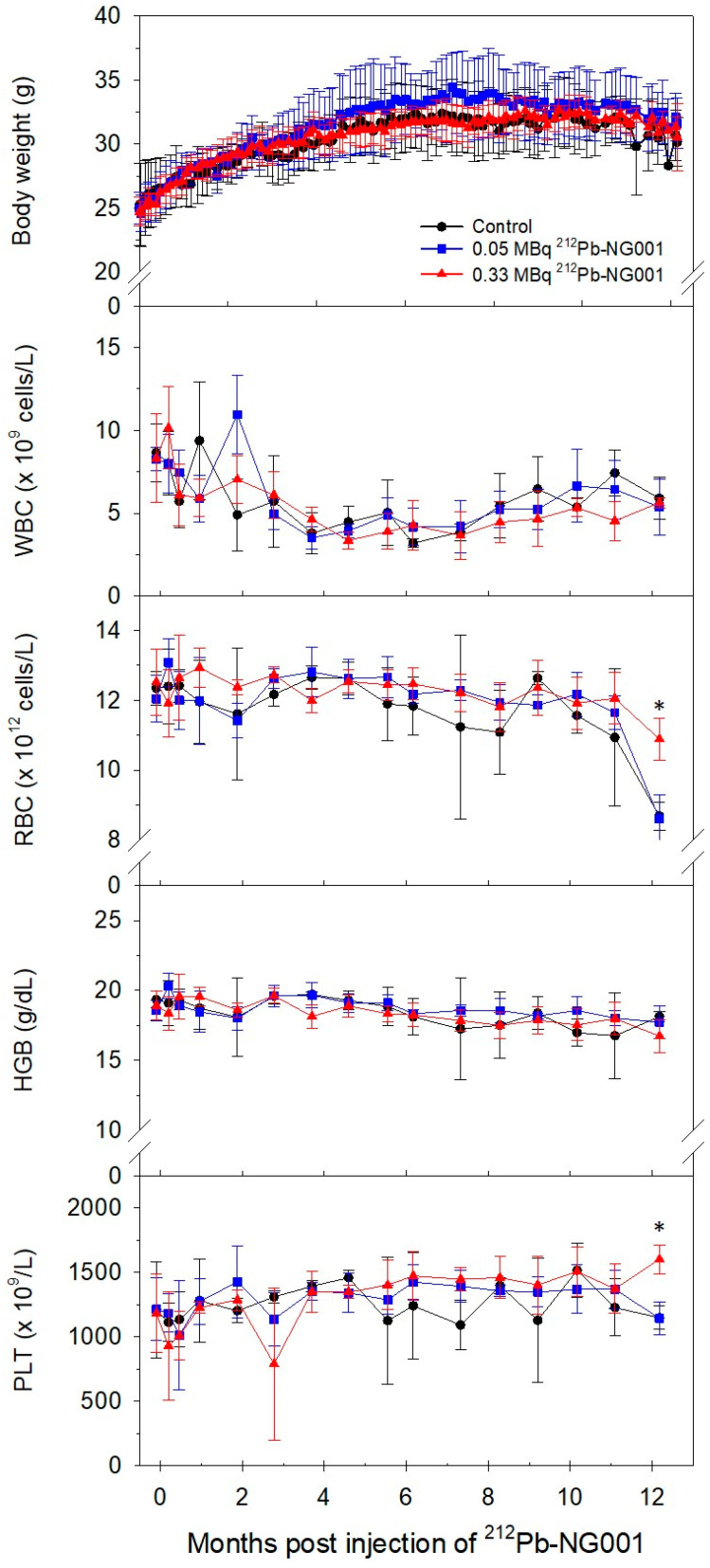
Changes in body weight and blood parameters, namely white blood cell (WBC), red blood cell (RBC), haemoglobin (HGB) and platelet (PLT) counts, of immunocompetent BALB/c mice after administration of saline (control, N = 3), 0.05 MBq (N = 5) or 0.33 MBq (N = 5) of ^212^Pb-NG001. * *p* < 0.05 compared to the other groups.

**Figure 5 ijms-22-04815-f005:**
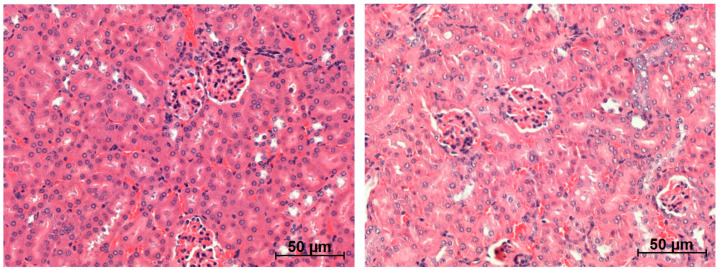
Representative histological images (×20 magnification) of kidneys of BALB/c mice 12 months after administration of saline (**left**) or 0.33 MBq of ^212^Pb-NG001 (**right**).

**Figure 6 ijms-22-04815-f006:**
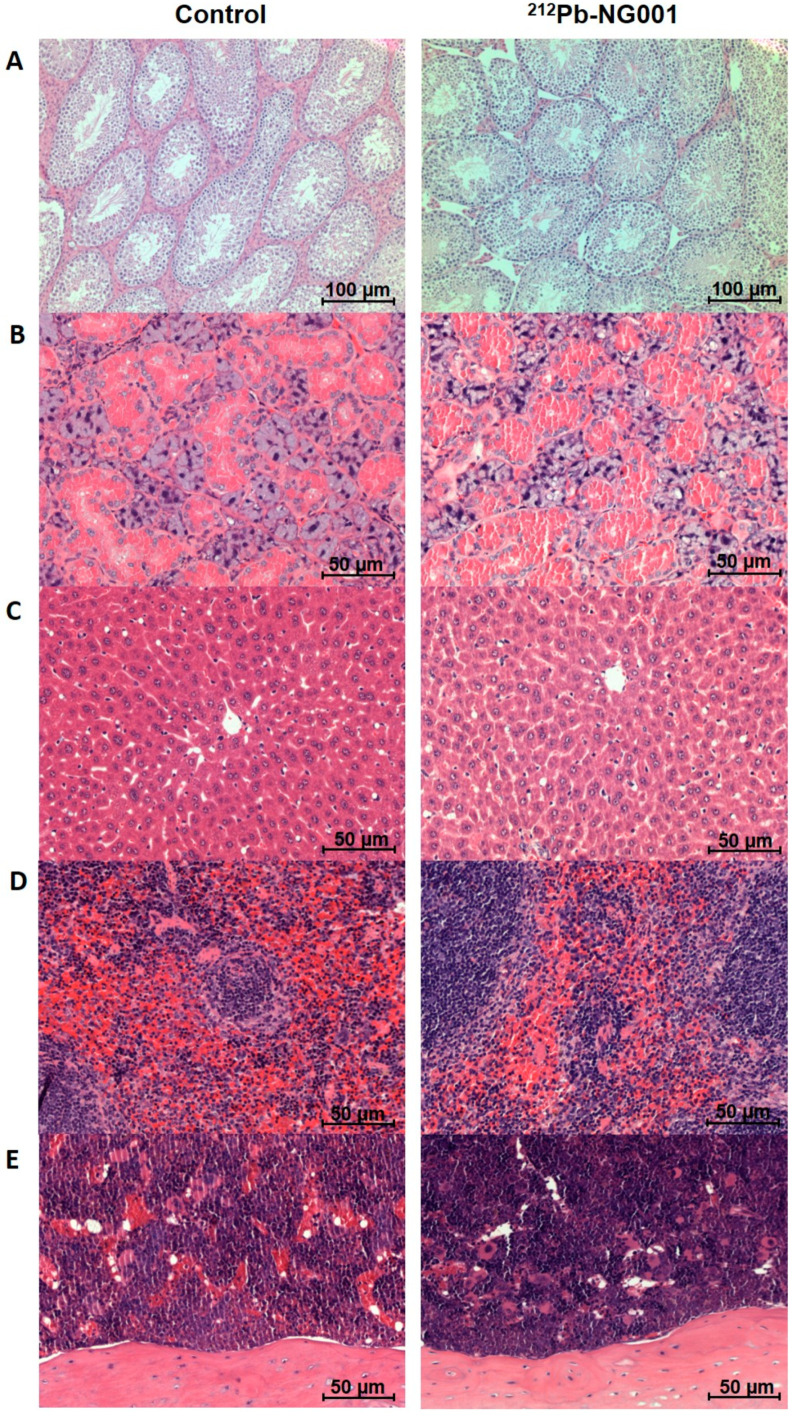
Representative histological images of testes (×10 magnification, (**A**)), salivary glands (×20 magnification, (**B**)), liver (×20 magnification, (**C**)), spleen (×20 magnification, (**D**)) and femur (×20 magnification, (**E**)) of BALB/c mice 12 months after administration of saline (left) or 0.33 MBq of ^212^Pb-NG001 (right).

**Figure 7 ijms-22-04815-f007:**
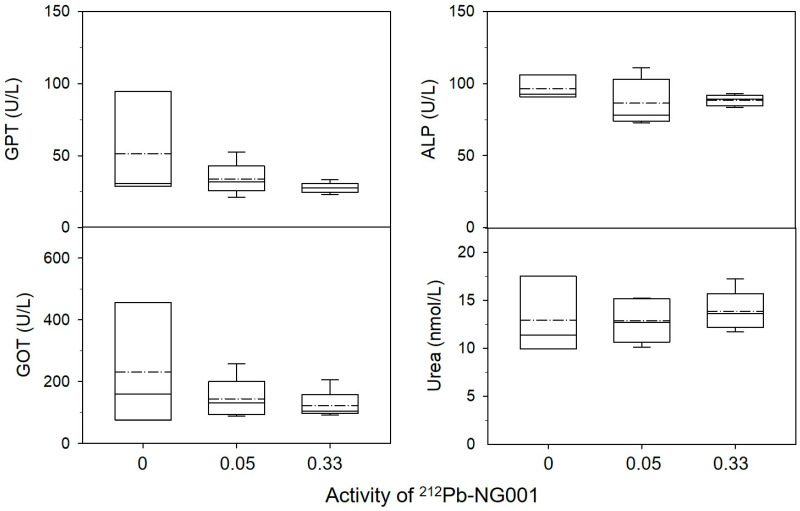
Blood chemistry in BALB/c mice treated with saline (N = 3), 0.05 MBq (N = 5) or 0.33 MBq (N = 5) of ^212^Pb-NG001. Confirmed normal values of bilirubin <8.55 µmol/L and creatinine <44.2 µmol/L were measured for all mice (results not shown). Bottom of the boxes represent the 25th percentile, dashed lines represent mean, solid lines represent median, top of the boxes represent the 75th percentile and whiskers represent the 5th and 95th percentiles. GOT, glutamic oxaloacetic transaminase; GPT, glutamic pyruvic transaminase; ALP, alkaline phosphatase.

**Table 1 ijms-22-04815-t001:** Biological and effective half-lives (T) of ^212^Pb-NG001 in selected organs/tissues in nude mice with C4-2 xenografts, N = 3–11.

Organ/Tissue	Pharmacokinetic Parameter	^212^Pb-NG001
Blood	$T_{\mathrm{effective}}$ $T_{\mathrm{biological}}$	4.2 ± 0.4 min 4.2 ± 0.4 min
Tumour	$T_{\mathrm{effective}}$ $T_{\mathrm{biological}}$	5.3 ± 0.9 h 10.6 ± 2.0 h
Kidneys	$T_{\mathrm{effective}}$ $T_{\mathrm{biological}}$	56.4 ± 5.4 min 52.2 ± 4.8 min

**Table 2 ijms-22-04815-t002:** The therapeutic effect of ^212^Pb-NG001 in athymic nude mice with C4-2 xenografts. The therapeutic index (TI) was calculated from median survival and median time for tumours to reach a volume of 1.5 cm^3^ in the treatment groups compared to the control group.

Treatment Group	Number of Mice	Median Survival (Days)	Median Time for Tumour to Reach 1.5 cm^3^ (Days)	TI, Median Survival (*p* Value)	TI, Tumour Volume (*p* Value)
Study 1					
Control	7	23	17.0		
0.25 MBq ^212^Pb-NG001	8	35	30.5	1.5 (<0.001)	1.8 (0.002)
Study 2					
Control	8	16.5	15.0		
0.30 MBq ^212^Pb-NG001	8	38.5	37.5	2.3 (0.012)	2.5 (0.015)
Study 3					
Control	6	27	25.5		
0.40 MBq ^212^Pb-NG001	6	74	64.0	2.7 (<0.001)	2.5 (0.002)

**Table 3 ijms-22-04815-t003:** Tumour and kidney uptake of various radiolabelled PSMA-binding ligands after 1, 4 and 24 h. N/A, not available.

Radioligand	Mouse Strain (Cell Line)	Tumour Uptake (%ID/G)			Kidney Uptake (%ID/G)			Reference
		1h	4h	24h	1h	4h	24h	
^212^Pb-NG001	Hsd: athymic nude-Foxn1^nu^ (C4-2)	23.3 ± 8.7	13.6 ± 2.1	11.3 ± 3.4	62.1 ± 7.0	9.6 ± 2.5	5.2 ± 0.8	Current study
^203^Pb-CA012	BALB/c nu/nu (C4-2)	8.4 ± 3.7	7.8 ± 0.9	3.3 ± 1.6	5.1 ± 2.5	1.6 ± 0.3	0.9 ± 0.1	Dos Santos et al., 2019 [5]
^203^Pb-L2	NOD-SCID gamma (PC-3 PIP)	22.5 ± 8.1	11.6 ± 4.2	8.5 ± 2.1	23.0 ± 11.9	3.8 ± 1.0	3.1 ± 0.8	Banerjee et al., 2020 [24]
^177^Lu-PSMA-617	NOD-SCID gamma (LNCaP)	15.1 ± 5.6	14.5 ± 1.8	10.9 ± 3.3	97.2 ± 19.4	26.6 ± 19.1	0.6 ± 0.2	Kuo et al., 2018 [30]
^177^Lu-PSMA-617	BALB/c nu/nu (LNCaP)	11.2 ± 4.2	N/A	10.6 ± 4.5	137.2 ± 77.8	N/A	2.1 ± 1.4	Benesova et al., 2015 [31]
^177^Lu-PSMA-617	BALB/c nu/nu (LNCaP)	N/A	14.4 ± 1.1	4.6 ± 0.6	N/A	14.1 ± 3.1	0.7 ± 0.1	Kelly et al., 2018 [32]
^177^Lu-PSMA-617	Athymic nude BALB/c (PC-3 PIP)	44.2 ± 12.0	56.0 ± 8.0	37.3 ± 5.8	9.8 ± 1.4	3.4 ± 3.2	0.8 ± 0.9	Benesova et al., 2018 [33]

**Table 4 ijms-22-04815-t004:** Therapeutic effect of various radiolabelled PSMA ligands in different subcutaneous animal models. Appendix A provides details of the mouse strains and cell lines used and specific activities in each individual study. The therapeutic index (TI) is calculated as the median survival of the treatment group divided by that of the control group.

Radioligand	Injected Activity Range	Therapeutic Index Range	References
^212^Pb-NG001	0.25–0.40 MBq	1.5–2.7	Current study
^212^Pb-L2	1.5–3.7 MBq	1.9–3.0	[24]
^177^Lu-PSMA-617	2–111 MBq	1.7–4.1	[3,30,36,58,59,60]
^225^Ac-PSMA-617	0.02–0.1 MBq	1.1–6.7	[59,61]

**Table 5 ijms-22-04815-t005:** Overview of the properties of relevant therapeutic radionuclides [65]. N/A, not available.

Radionuclide	Total Energy Emitted Per Decay (MeV)	Preclinical Activity Injected in Mouse Model *	Relevant Clinical Doses
^212^Pb (t_1/2_ = 10.6 h)	7.90 (1 α, 2 β)	0.3 MBq	60–80 MBq **
^177^Lu (t_1/2_ = 6.7 days)	0.15 (1 β)	30 MBq	6–8 GBq [6]
^225^Ac (t_1/2_ = 10.0 days)	27.90 (4 α, 2 β)	30 kBq	6–8 MBq [12,13,14]
^213^Bi (t_1/2_ = 45.6 min)	8.47 (1 α, 2 β)	N/A	296 MBq [17]

* Activity dose that result in TIs of around 2–3 (compared to control), ** calculated from a factor of 200–267 compared to mice, corresponding to mouse–human factor of 200–267 for ^177^Lu and ^225^Ac.

## Data Availability

The data presented in the study are available in the Appendix A or on request from the corresponding author.

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
