# Peer review of "Evaluation of the PSMA-Binding Ligand 212Pb-NG001 in Multicellular Tumour Spheroid and Mouse Models of Prostate Cancer"

_ijms, 2021, doi:10.3390/ijms22094815_

Round 1
Reviewer 1 Report
The manuscript entitled “Evaluation of the PSMA-binding ligand 212Pb-NG001 in multi- 2 cellular tumour spheroid and mouse models of prostate cancer” by Stenberg et al. contains a well-planned study evaluating 212Pb-NG001 as a radioligand for PSMA targeting. This is an interesting paper and the work presented is suitable for publication in the International Journal of Molecular Sciences. However, the paper is too long. I believe the paper will be more useful and readable if it is reduced at least by 20 to 30%. For example, the experimental section can be reduced significantly by referring to J. of Labelled Compounds and Radiopharmaceuticals paper that was published in 2020. Similarly, the results and discussion section will be more effective if these are presented in significantly shortened form.
Author Response
Comment:
The manuscript entitled “Evaluation of the PSMA-binding ligand 212Pb-NG001 in multi-2 cellular tumour spheroid and mouse models of prostate cancer” by Stenberg et al. contains a well-planned study evaluating 212Pb-NG001 as a radioligand for PSMA targeting. This is an interesting paper and the work presented is suitable for publication in the International Journal of Molecular Sciences.
Answer:
Thank you for this positive feedback.
Comment:
However, the paper is too long. I believe the paper will be more useful and readable if it is reduced at least by 20 to 30%. For example, the experimental section can be reduced significantly by referring to J. of Labelled Compounds and Radiopharmaceuticals paper that was published in 2020. Similarly, the results and discussion section will be more effective if these are presented in significantly shortened form.
Answer:
We have now shortened the experimental section considerably (removed around 300/1700 words) by referring to and citing our previous papers. We have also tried to shorten the results and discussion sections but found it difficult without removing important information. In addition, Reviewer 2 suggested to add more details on kidney toxicity prevention in the discussion.
We have corrected grammatical mistakes and have had an English native speaker to edit the manuscript.

Reviewer 2 Report
See the attached file for the comments

Author Response
Reviewer 2
Comments and Suggestions for Authors
Comment:
The manuscript titled "Evaluation of PSMA 212Pb-NG001 binding ligand in multicellular tumor spheroid and prostate cancer mouse models" describes a PSMA targeted ligand labeled with 212Pb for the treatment of prostate cancer. The Authors clearly demonstrate the efficacy of this new therapeutic approach in vitro on the spheroid model resembling avascular tumor micrometastases and in vivo on subcutaneous tumors grown in nude mice. Furthermore, the Authors point out the absence of toxicity after their treatments. The topic covered in this manuscript is very interesting because there are currently no effective therapies available for the management of metastatic and castration-resistant prostate cancer. I suggest this document needs minor revisions before publication.
Answer:
Thank you for this positive feedback.
Comment:
Several issues need to be clarified: 1) I think that the Authors should add to the text not only the diameter of the spheroids but also the volume of the spheroids at time zero (i.e. at the beginning of the treatments). These data could improve understanding of the effects of treatments in reducing growth. Furthermore, the Authors verified whether the spheroids treated with higher doses are sterilized at the end of the experiments or can they grow back? If they grow back are they sensitive to a second challenge with the same drug?
Answer:
The volume of the spheroids at time zero has been added to the result section (page 2, lines 80-81): “Each spheroid was formed from 500 cells and reached a diameter of 400±60 µm (volume of 26±3 µm3) after 5 days in culture (day of treatment, day 0)”.
The spheroids treated at the highest doses were followed only for 20 days. During this period, we did not observe any regrowth. At the end of the study, spheroids were stained with fluorescein diacetate (FDA, marker for viable cells) and propidium iodide (PI, marker for dead cells). Results showed that some of the C4-2 cells were viable in the spheroids treated with the highest activities. This means that the spheroids can start to re-grow. We have not performed spheroid experiments with a second treatment, but this is interesting and will be discussed for further studies.
Comment:
2) How do Authors explain the non-accumulation of 212Pb-NG001 in the salivary glands of injected mice? Looking at the structure of the ligand part of this PSMA-radioligand it appears to be similar to PSMA-617.
Answer:
Salivary gland uptake of 177Lu-PSMA-617 in mice has also previously been found to be very low (<0.5%ID/g after 1h, Benesova et al., 2018, Banerjee et al., 2020 and Stenberg et al., 2020). In contrast, accumulation of PSMA ligands in human salivary glands is demonstrated to be high, partly described by a higher PSMA level (Roy et al., 2020) but also PSMA unrelated uptake mechanisms (Rupp et al., 2019). The details of the mechanisms are currently unclear.
Comment:
3) Do the authors have data on tumor biodistribution in mice when an excess of cold NG001 is administered to highlight the specificity of tumor accumulation in vivo?
Answer:
Tumor biodistribution in mice after pre-block with cold NG001 have not yet been done but are planned in the near future. Until now, we have only used the PSMA-negative cell line PC-3 Flu as a negative control to verify tumor specificity.
Comment:
4) I partially agree with the Authors’ statement “In a product of higher specific activity, a larger radionuclide dose could theoretically be bound to the tumour cells before binding saturation but it may also cause added toxicity to normal tissues with modest expression of PSMA” (see page 11, rows 229-231), because increasing the amount of binding reagent (i.e. in the case of administration of a radioligand with lower specific activity) also increases the possibility of targeting cells expressing lower PSMA. What do you think Authors?
Answer:
We agree with the reviewer that the statement was unclear. Increasing the specific activity elevates the level of PSMA-targeted radioligand bound to the cells expressing PSMA (Boas et al., 2020). Reducing the specific activity by adding unlabeled ligand or another molecule blocking the radioligand binding (Banerjee et al., 2020), can reduce the initial kidney uptake of the radioligand. However, low specific activity can also cause lower tumour to kidney ratio at later time points (Fendler et al., 2017). For radionuclides with relatively short half-life, it is favorable to avoid a too high initial kidney uptake by moderately restricting the specific activity as long as the tumour uptake is maintained.
However, since Reviewer 1 suggested to shorten the discussion section, we have removed the statement and will not include this quite complicated discussion on specific activity in the text of our revised manuscript.
Comment:
5) Authors stated “However, kidney toxicity may be a potential problem in clinical use of small molecular PSMA radioligands and should be carefully monitored.” (see page 12, rows 295-296) Could the Authors briefly summarize the strategy currently used in clinics to decrease this renal accumulation in order to avoid kidney damages?
Answer:
Strategies currently used to reduce kidney toxicity of TRT is added to the discussion section (page 13, lines 300-305): “The standard strategy to avoid kidney damage in clinical targeted radionuclide therapy are to ensure that the patient is well hydrated and to co-infuse the basic amino acids lysine and arginine, to reduce renal retention of the radioligands [41, 46, 47]. In addition, several strategies to reduce kidney toxicity is currently under investigation, including dose fractionation and co-injection with cold PSMA ligand, succinylated gelatin or radioprotectors, such as α1-microglobulin or amifostine [41, 48-53].”
References 46-53 were added, and the later reference numbers were changed.